# Variations and Depth of Formation of Submesoscale Eddy Structures in Satellite Ocean Color Data in the Southwestern Region of the Peter the Great Bay

Nadezhda A. Lipinskaya [†] , Pavel A. Salyuk *,[†] and Irina A. Golik

V.I. Il'ichev Pacific Oceanological Institute, Far Eastern Branch, Russian Academy of Sciences (POI FEB RAS), 690041 Vladivostok, Russia; lipinskaya.na@poi.dvo.ru (N.A.L.); lastovskaya@poi.dvo.ru (I.A.G.)
* Correspondence: psalyuk@poi.dvo.ru
[†] These authors contributed equally to this work.

**Abstract:** The aim of this study was to develop methods for determining the most significant contrasts in satellite ocean color data arising in the presence of a submesoscale eddy structure, as well as to determine the corresponding depths of the upper layer of the sea where these contrasts are formed. The research was carried out on the example of the chain of submesoscale eddies identified in the Tumen River water transport area in the Japan/East Sea. MODIS Aqua/Terra satellite data of the remotely sensed reflectance (*Rrs*) and *Rrs* band ratio at various wavelengths, chlorophyll-a concentration, and, for comparison, sea surface temperature (*sst*) were analyzed. Additionally, the results of ship surveys in September 2009 were used to study the influence of eddy vertical structure on the obtained remote characteristics. The best characteristic for detecting the studied eddies in satellite ocean color data was the MODIS *chlor_a* standard product, which is an estimate of chlorophyll-a concentration obtained by a combination of the three-band reflectance difference algorithm (CI) for low concentrations and the band-ratio algorithm (OCx) for high concentrations. At the same time, the weakest contrasts were in *sst* data due to similar water heating inside and outside the eddies. The best eddy contrast-to-noise ratio according to Rrs spectra is achieved at 547 nm in the spectral region of seawater with maximum transparency and low relative errors of measurements. The Rrs at 678 nm and associated products may be a significant characteristic for eddy detection if there are many phytoplankton in the eddy waters. The maximum depth of the remotely sensed contrast formation of the considered eddy vertical structure was ~6 m, which was significantly less than the maximum spectral penetration depth of solar radiation for remote sensing, which was in the 14–17 m range. The results obtained can be used to determine the characteristics that provide the best contrast for detecting eddy structures in remotely sensed reflectance data and to improve the interpretation of remote spectral ocean color data in the areas of eddies activity.

**Keywords:** submesoscale; eddies; chlorophyll-a; remotely sensed reflectance spectra; sea surface temperature; satellite; contrast; contrast-to-noise; light penetration depth; vertical structure

## 1. Introduction

Submesoscale eddies are among the most common but poorly understood forms of the eddy motion of oceanic waters. In such eddies, the centrifugal force can exceed the Coriolis force in magnitude, leading to high vertical velocities in their cores (10–100 m/day), small horizontal sizes from 100 m to 10 km, and lifetimes from a few hours to a few days [1,2]. Submesoscale eddies are an important factor in the functioning of marine ecosystems [3,4], the transport and mixing of substances [5,6], and heat fluxes [7,8]. In addition, accurately quantifying submesoscale eddies can be helpful in increasing the accuracy of ocean applications, such as oil spills [9] and iceberg detection [10].

One of the most common methods for studying submesoscale eddies in the ocean is satellite measurements. In this case, it is very important to have methods for detecting

eddies in all possible spectral bands, which will increase the spatial and temporal coverage of the studied phenomenon. This is especially relevant for submesoscale eddies, as these are highly dynamic structures with small sizes and small lifetimes.

Traditionally, satellite radar data have been used to study marine eddy structures [11–13], even in the presence of clouds. Especially useful for detecting submesoscale eddies are synthetic aperture radar (SAR) data [14,15]. Additionally, the use of satellite data in visible and infrared (IR) spectral bands significantly enhances the potential for studying submesoscale eddies [16,17]. The multi-sensor approach allows for increased spatial and temporal coverage of the phenomena under investigation, providing more information to determine the mechanisms of eddies' formation and the composition of substances involved in vortex motion. But, in this paper, the study focuses on the investigation of eddy manifestations in the visible and IR spectral ranges and does not concern the use of the microwave range. This is due to the fact that, first, there are a number of problems associated with satellite data atmospheric correction errors in the optical range that still need to be solved in terms of selecting optimal characteristics for eddy detection [18,19], and second, remotely sensed data in the visible spectral range are affected by changes in the seawater vertical structure of optically active substances [20], which may serve as an additional factor for analyzing marine eddies.

In the satellite IR images, eddies are distinguished due to sea surface temperature contrasts, which is a basic oceanographic parameter [21,22]. However, for some areas, in a certain season, the manifestations of eddies in thermal contrasts may be weak or entirely absent [23]. In cases when satellite IR sensing fails to identify eddies due to small contrasts in the sea surface temperature fields, data in the visible range can be used, where the contrast can be significant due to different concentrations of chlorophyll-a (chl-a) [23–25], suspended solids (SSs) [26], colored dissolved organic matter (CDOM) [27], and even due to the distribution of broken ice [28]. These characteristics from satellite ocean color data are calculated from measurements of remotely sensed reflectance (*Rrs*) and can be attributed to optical characteristics of seawater. Despite the fact that the values of satellite chl-a concentrations may contain significant systematic errors, both due to atmospheric correction errors and non-standard SS or CDOM contributions to *Rrs* spectra [29,30], the structures of hydro-physical processes will still be manifested in the chl-a concentration fields. Moreover, in the case of the influence of SS or CDOM on the quality of satellite chl-a concentration estimation, it can only enhance the visible contrasts.

The advantage of remotely observing eddies based on thermal contrasts is that the measured differences are a result of varying water masses with different temperatures, rather than differences in optically active seawater substances that can be transported, deposited, or influenced by the functioning of phytoplankton communities [31]. In addition, in the case of analyzing optical images in the visible spectral range, the manifestation of different water masses from different depths may be affected [32], which may also lead to the blurring of contrasts. These factors should be considered when interpreting remote sensing results.

It is important to note that the availability of high-quality satellite images in the optical range is not consistent from day to day due to cloud cover variability, both in the visible and IR parts of the spectrum. In this regard, it is not always possible to track the trajectories of movement and the number of eddies in the sea [23]. This is especially aggravated by the fact that the study of submesoscale eddies mainly requires high spatial resolution satellite data (pixels less than 100 m), which are not available for every day. However, medium spatial resolution data (500–1000 m) may be suitable if the investigated eddies are larger than 5000 m.

Thus, to conduct a comprehensive study of submesoscale eddies using satellite data in the optical range, it is crucial to utilize both ocean color and temperature data. Additionally, it is necessary to estimate the depths at which these phenomena can be observed in remotely sensed data. This will increase the spatial and temporal coverage of the

analyzed phenomena and provide an opportunity for additional interpretation of the obtained results.

The purpose of this work is to determine remotely estimated parameters from the visible and IR ranges of electromagnetic radiation, by which it is possible to obtain the most significant contrasts for the study of submesoscale eddies, as well as to determine the depth at which the presence of the considered eddy structure is not visible on satellite images of ocean color. This study was carried out on the example of submesoscale eddies detected in the southwestern part of the Peter the Great Bay in the area of influence of the Tumen River waters during a series of coastal expeditions in September 2009.

Previous studies on this topic typically focus on analyzing the amplitudes of changes in characteristics inside and outside of eddies [23,27,32], without considering their significance in relation to statistical noise. The knowledge of contrast/noise characteristics is necessary to investigate the quality of detection of submesoscale eddies in satellite images.

The hypothesis to be tested is that the most effective characteristics for detecting submesoscale eddies may include different optical contrasts obtained at various wavelengths in the visible range of the spectrum, as well as temperature contrasts obtained in the IR range of the spectrum. The effectiveness of these characteristics may depend on the hydro-physical conditions and the vertical structure of optically active substances in the water column.

The paper does not consider gradients of magnitudes and all other possible mathematical methods of contrast enhancement, because they can be applied to any of the selected characteristics. The question is which optical characteristic to take as a basis for calculating contrasts from the point of view of the physics of processes. Further, the selected contrast characteristics can be enhanced or analyzed by all kinds of image recognition methods.

## 2. Data and Methods

### 2.1. Study Area

The studies were carried out in the southwestern part of the Peter the Great Bay (Figure 1) in the area of active eddy generation [21,33], where the formation of submesoscale eddies is occasionally observed, carrying Tumen River waters in the northeastern direction [6,34]. This determines the relevance and importance of research in this region, because the Tumen River is one of the major rivers of the Japan/East Sea and may be a source of pollutants [35–39], or, at least, it may bring freshened waters with increased content of CDOM and SS into marine ecosystems. This is considered as one of the causes of hypoxia in the study area [40,41] and as a potential factor impacting the ecosystems of the Far Eastern Marine Reserve [6].

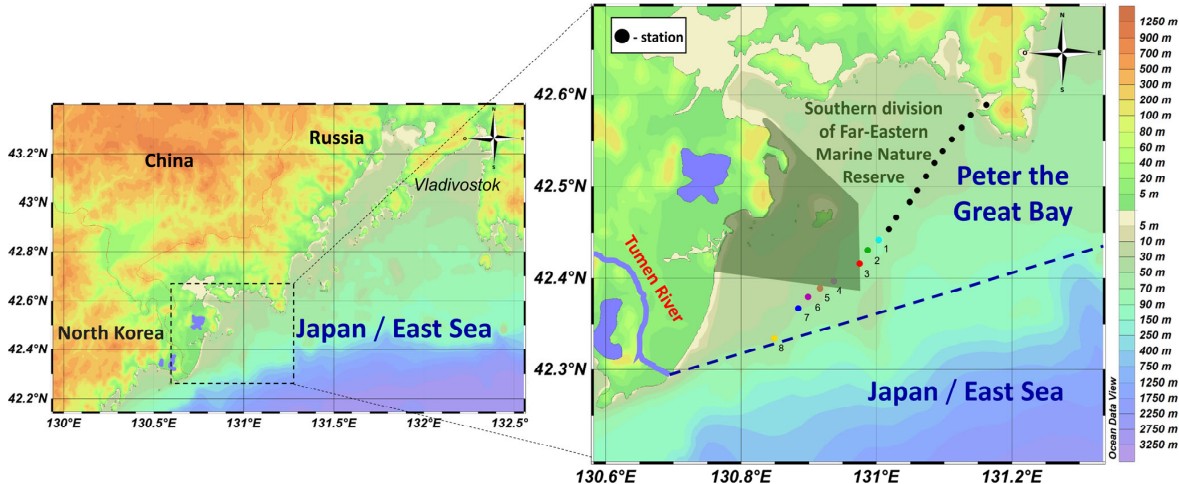

**Figure 1.** Study area. Big dots indicate ship measurements made on 4 September 2009. The numbers near the dots correspond to the station numbers used later in the text. Green polygon indicates the southern area of the Far Eastern Marine Reserve.

*2.2. Satellite Data*

Level 2 data from the MODIS-Terra/Aqua satellite color scanners were utilized for the comparative analysis of submesoscale eddy manifestations. Optical characteristics from the visible spectral range and sea surface temperature (*sst*) were used in the analysis. The pixel size in the satellite images was within ≈1–2 km and depended on the viewing angle and spectral channels used on the satellite radiometer.

The *sst* satellite data were obtained from measurements in the long-wave thermal infrared spectral interval (10–12 μm) using standard algorithms [42], implemented in MODIS Reprocessing R2019.0.

The analyzed optical characteristics included satellite estimates of chl-a concentration (*chlor_a$_{sat}$*) based on the algorithm of Hu et al. [43], using a combination of the empirical band difference approach (CI) at low chorophyll concentrations [18,43] with a band ratio approach (OCx) at higher chlorophyll concentrations [44]; remotely sensed reflectance was measured at all MODIS satellite radiometer wavelengths (*Rrs$_{sat}$*($\lambda$)) in the visible range: 412, 443, 469, 488, 531, 547, 555, 645, 667, and 678 nm; and the band ratios at the same wavelengths (*BR$_{sat}$*($\lambda$)). In general, regardless of the data source, the presented characteristics were calculated according to the following formulas:

$$log_{10}(chlor\_a_{CI}) = a_{0CI} + a_{1CI}\left(Rrs(555) - \left[Rrs(443) + \frac{555-443}{667-443} \times (Rrs(667) - Rrs(443))\right]\right), \tag{1}$$

$$log_{10}(chlor\_a_{OCx}) = a_0 + \sum_{i=1}^{4} a_i log_{10}\left(\frac{max(Rrs(443), Rrs(488))}{Rrs(547)}\right), \tag{2}$$

$$chlor\_a = \begin{cases} chlor\_a_{CI} \; if \; chlor\_a_{CI} \leq 0.25, \\ \frac{chlor\_a_{CI}(0.35 - chlor\_a_{CI})}{0.35 - 0.25} + \frac{chlor\_a_{OCx}(chlor\_a_{CI} - 0.25)}{0.35 - 0.25} \; if \; 0.25 < chlor\_a_{CI} \leq 0.35, \\ chlor\_a_{OCx} \; if \; chlor\_a_{CI} > 0.35, \end{cases} \tag{3}$$

$$BR(\lambda) = Rrs((\lambda)/Rrs(555), \tag{4}$$

where $a_{CI}$ = {−0.4287, 230.47} and ai = {0.26294; −2.64669; 1.28364; 1.08209; −1.76828} in accordance with [45] for MODIS Reprocessing 2022.0.

The studies were conducted using MODIS-Aqua/Terra satellite data for 31 August, 1 September, 2 September, and 4 September 2009 (Figure 2). During this period, the formation of a series of submesoscale eddies with sizes ranging from 3 to 10 km was observed [6]. The movement of these eddies in the northeastern direction was determined. On 1–2 September 2009, several high-quality satellite images were obtained, which revealed that the investigated eddies were cyclonic in nature [6].

The submesoscale eddies were identified through expert evaluation based on the analysis of the chl-a concentration images and chl-a concentration gradients obtained using the Sobel operator. Ellipsoidal structures were expected to be visible in the field of gradient values, while distinct spots were expected to appear on the chl-a concentration maps. The boundaries of the eddies were determined by approximating the maximum gradient values with an ellipse. Each eddy was assigned an identifier (ID$_{DD.NN}$), where DD represented the ordinal number of the investigated day and NN represented the eddy's number on that day. A total of 11 eddy structures, indicated in Figure 2, were analyzed during the study period.

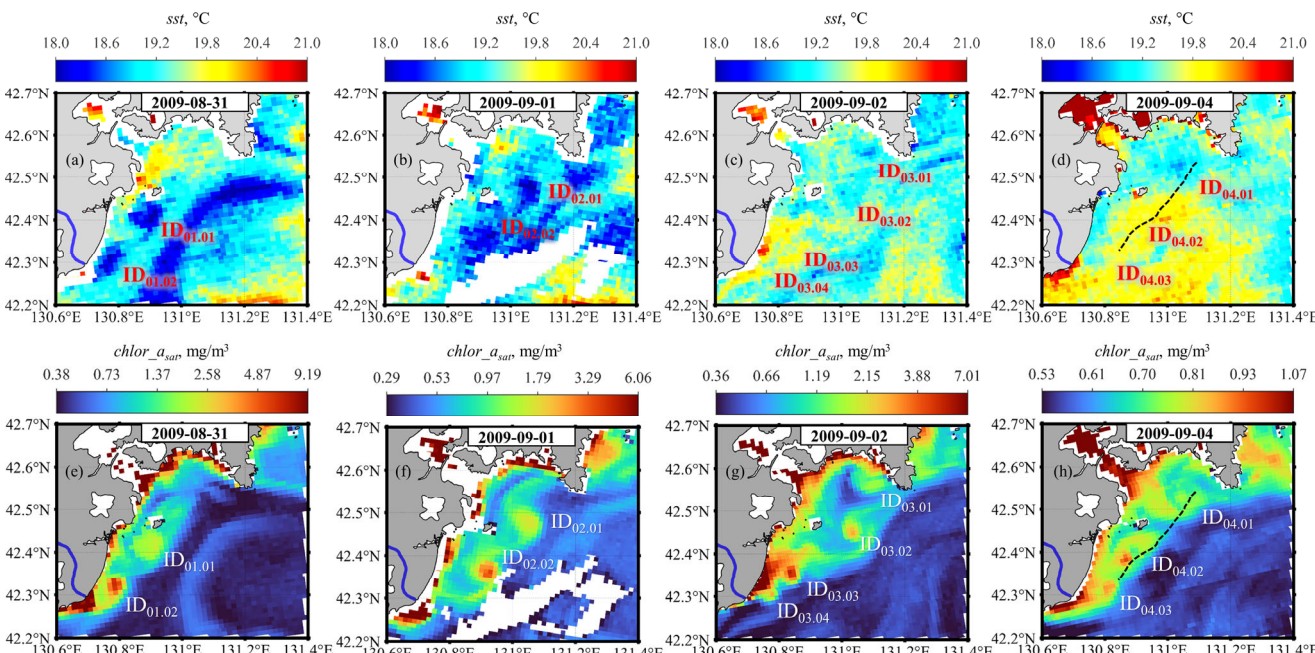

**Figure 2.** MODIS-Aqua/Terra satellite images in the southwestern part of the Peter the Great Bay from 31 August 2009 to 4 September 2009. Top row (**a**–**d**)—*sst* data. Bottom row (**e**–**h**)—*chlor_a$_{sat}$* data. The identifiers denote the eddies under consideration.

### 2.3. In Situ Data

To validate the identification of eddies in satellite images, the satellite data of *sst* and *chlor_a$_{sat}$* were compared with spatial data from shipboard flow measurements of temperature ($T_{flow}$) and chl-a concentration ($Chl_{flow}$). These measurements were obtained on 4 September 2009, for eddy ID$_{04.02}$. The corresponding track of shipboard measurements is shown by the dashed line in Figure 2d,h. The measurements were obtained as part of a series of coastal cruises using an SBE-45 flowing thermosalinograph and the flowing laser fluorimeter [46]. The fluorimeter provides the measurements of seawater fluorescence spectra in the range of 560–730 nm under excitation by 532 nm radiation, from which chl-a fluorescence intensity was calculated [46].

To analyze the internal structure of eddy ID$_{04.02}$, we used in situ data from CTD and bio-optical vertical measurements, as well as shipboard measurements of the remote sensing reflectance coefficients ($Rrs_{ship}$). The data were collected both inside and outside the eddy ID$_{04.02}$. The schematic location of the stations relative to the eddy structure is presented in Figure 3.

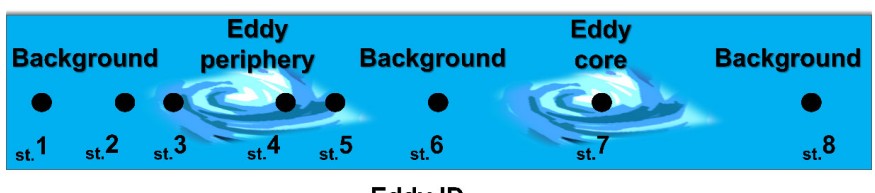

**Figure 3.** Scheme of oceanographic stations along the eddy structure ID$_{04.02}$ on 4 September 2009, where data from remote sensing and in situ ship measurements were collected. The ship route followed a direction from northeast to southwest. The numbers correspond to the ship's station numbers.

In situ vertical profiles of chl-a ($Chl_{insitu}(z)$) and CDOM ($CDOM_{insitu}(z)$) concentrations were obtained at stations 1–8 using a SeaBird SBE 19plus submersible hydrologic CTD probe with calibrated WetLabs WetStar fluorometers. The excitation central wavelengths

used in the WetStar sensors were ~460 nm for chl-a and ~370 nm for CDOM, while the registered central wavelengths were ~690 and ~460 nm, respectively. Calibration of chl-a fluorescence measurements was performed based on a comparison with standard extract spectrophotometric determinations of chl-a concentrations in seawater samples. Calibration of CDOM fluorescence intensity was performed before the expedition in μg/L of quinine sulfate dihydrate (QSUs—quinine sulfate units).

The $Rrs_{ship}(\lambda)$ spectra were obtained from a ship by an ASD FieldSpec Hand Held hyperspectral radiometer according to the methodology from NASA protocols [47]. The methodology was adapted to the specific conditions of using the radiometer [48]. The advantage of above-surface ship measurements is the absence of the influence of the atmosphere on the registered signal. Thus, the values obtained are determined mainly by the seawater optical characteristics and their vertical profiles in the water column.

### 2.4. Determination of Contrasts and Noise in Satellite Images

For each detected submesoscale eddy, the values of *sst*, $chlor\_a_{sat}$, $Rrs_{sat}(\lambda)$, and $BR_{sat}(\lambda)$ inside the eddy structure and in four zones around the eddy core (front, back, right, and left relative to the eddy motion path) were selected for analysis, as shown in the scheme in Figure 4. The boundaries of the "in" zone were determined from the location of the largest $chlor\_a_{sat}$ gradient values around the eddy core by approximation with an ellipse. Smoothing techniques were applied to all analyzed datasets using two methods: (sm1) two-dimensional median filtering and Gaussian filter with medium smoothing to identify the local extremum inside the eddy; (sm2) two-dimensional median filtering and Gaussian filter with strong smoothing to identify local extrema in the region surrounding the eddy. The median filtering helped remove salt-and-pepper-type noise, while Gaussian smoothing reduced statistical noise. The parameters for sm1 smoothing were selected to prevent underestimation of the absolute value of the extremum within the eddy structure. Stronger sm2 smoothing was used to remove the influence of unusual intrusions involved in vorticity in the region around the eddy.

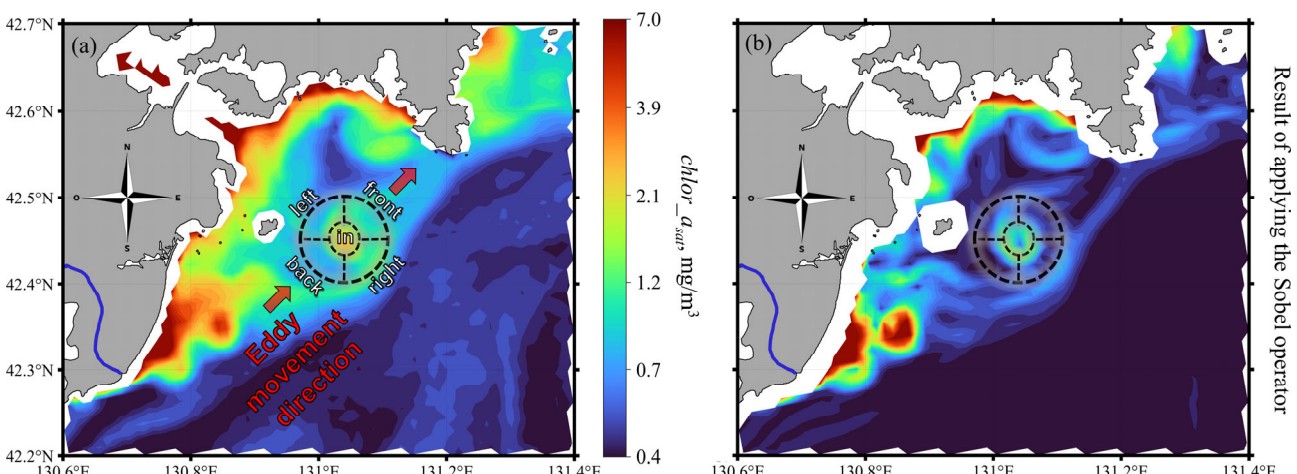

**Figure 4.** Schematic illustrating the selection of regions inside and outside the eddy for calculating the contrast-to-noise ratio (*CNR*) on the example of eddy ID$_{03.02}$: (**a**) initial $chlor\_a_{sat}$ concentration data; (**b**) the result after applying the Sobel operator.

For each eddy, the contrast-to-noise ratio (*CNR*) was calculated by dividing the difference between the extreme (minimum or maximum) smoothed values of the satellite data inside and outside the eddy by the noise of the satellite image:

$$signal = data_{sm1}(inside), \tag{5}$$

$$background = data_{sm2}(outside), \tag{6}$$

$$noise = min\left(noise_{signal}, noise_{background}\right), \tag{7}$$

$$CNR = \frac{signal - background}{noise} = \frac{contrast}{noise}, \tag{8}$$

where *signal* is the maximum or minimum value of the smoothed satellite data according to method sm1 ($data_{sm1}$) *inside* the eddy; *background* is the minimum or maximum (opposite to the *signal*) value of more strongly smoothed satellite data according to method sm2 ($data_{sm2}$) in waters *outside* the eddy; *noise* is the noise of the satellite image determined in the regions of near-zero gradients in the area of the observed eddy; $noise_{signal}$ is the *noise* at the *signal* level; and $noise_{background}$ is the *noise* at the *background* level.

Additionally, relative noise values were calculated in order to perform a comparative analysis of noise characteristics between different spectral channels in the visible range, taking into account the level of the analyzed signals:

$$\%noise_{rel} = 100\% \times {}^{noise}/_{\min(signal, background)}, \tag{9}$$

In the worldwide scientific literature, parameters similar to the presented *CNR* are widely used to characterize the contrast (conspicuity) of structures in images measured by different instruments [49–51]. The differences consist in the methods of determining the difference between signals and image noise.

The *noise* of the satellite image in this article was determined using the method of local homogeneous blocks, based on the works [52,53]. The satellite image in the study area was divided into numerous small blocks of $4 \times 4$, $6 \times 6$, and $8 \times 8$ pixels. In our study, we used a modification of the noise estimation method, which consisted in filtering out blocks when large values of the gradient of the analyzed characteristic were observed. The gradient threshold value was set as twice the mode of the distribution of all gradient values obtained in the image. This condition filtered out the blocks that showed all kinds of inhomogeneities associated with eddies, fronts, and satellite image artifacts well. Within the remaining blocks, the local mean and local standard deviation (SD) values were calculated.

Next, a plot of the dependence of the square of the local SD on the square of the local mean was analyzed. If no significant linear dependence was observed, the noise was considered additive and its values were equated to the mode of distribution of all local SDs obtained in the image. Conversely, if a significant linear dependence between the square of the local SD and the square of the local mean was observed, the noise was considered multiplicative. The corresponding linear dependence was approximated by a straight line, and the SD level was determined based on either the signal level or the background level, depending on which level was lower. The corresponding linear relationship was approximated using a straight line, and the noise was determined based on the smaller of the SD values at either the signal level or the background level.

It is important to note that the statistical noise estimate obtained does not represent the true error of the satellite measurements, which may consist of various components and include all kinds of additional systematic errors. The statistical noise only takes into account variations in satellite data in homogeneous blocks in order to determine the possibility of detecting different structures on the image in the study area but not to estimate the true error of the analyzed parameters. The estimated noise value is the sum of two factors: signal fluctuations in the region not significantly affected by dynamic processes and the instrumental noise of satellite measurements.

The obtained *CNR* values according to the formula (Equation (8)) can be either negative or positive, depending on the satellite data used and the hydrodynamic process under consideration. The presence of an eddy can result in either an increase or a decrease in the analyzed characteristics. In order to retain information about the sign of the signal change in relation to the background, we did not use an absolute value function in the formula for calculating *CNR*.

The CNR, calculated using formula (Equation (8)), can be interpreted as the maximum contrast in the satellite data caused by the presence of an eddy structure, taking into account measurement errors. This value is necessary to estimate the possibility of detecting the eddy rather than the size of the corresponding structure. A rough estimate to determine if it is possible to visually detect an eddy in a satellite image can be expressed by the following condition:

$$|CNR| \geq 2, \tag{10}$$

and to detect the structure using numerical methods:

$$|CNR| \geq 1. \tag{11}$$

However, it should be noted that Equations (10) and (11) are not absolute rules, as the detection capability will also depend on the number of neighboring pixels that meet the presented conditions. When interpreting the results, the following factors should be taken into consideration:

1.  If there are numerous nearby pixels where the condition of $1 \leq |CNR| \leq 2$ is met, it is possible to visually perceive such a structure in the case of an appropriate color scale. The selection of an optimal color scale can enhance the visibility of the structure and aid in its identification.
2.  The same is applicable for values $|CNR| \leq 1$, if there are many such pixels, and they are located next to each other, then such a structure can be detected by modern methods of image recognition.
3.  And vice versa for scenarios (1) and (2): if there is only one pixel that satisfies the conditions $|CNR| \geq 1$ or even $|CNR| \geq 2$, it would be difficult to classify such a structure as an eddy. However, our approach to smoothing the satellite data for signal (Equation (5)) and background (Equation (6)) estimations avoids cases of random outliers of individual pixels near the eddy structure. There is an exception for situations where the size of the natural phenomenon is comparable to the size of a satellite pixel, but this is not the case in our paper.

Therefore, for the purpose of this paper, which is to compare satellite measurements and determine the best contrast characteristics, it will be sufficient to use the simple rules (Equations (10) and (11)) without considering the area or the number of pixels that meet these conditions.

### 2.5. Determination of the Maximum Depth of Remotely Sensed Contrast Formation

It is known that the main part of *Rrs* is formed in the solar light penetration layer for remote sensing ($Z_{90}$), which in the case of a homogeneous ocean is defined as follows [54]:

$$Z_{90}(\lambda) = 1/K_d(\lambda), \tag{12}$$

where $K_d(\lambda)$—the diffuse attenuation coefficient of downwelling irradiance. The $Z_{90}(\lambda)$ is the depth at which 90% of the scattered sea radiation at wavelength $\lambda$ (excluding specular reflection) is formed.

When dealing with a non-uniform vertical distribution of optically active substances in the near-surface sea layer, it is crucial to take into account that the upper layers have a higher optical contribution to the formation of upwelling sea radiation compared to the lower layers. An appropriate method to account for this effect was presented in [20] (Formula (24)), which can be modified to determine $Z_{90}(\lambda)$ for such nonhomogeneous ocean conditions:

$$\frac{R(\lambda, 0^-, Z_{90}(\lambda))}{R(\lambda, 0^-, \infty)} = 0.9, \tag{13}$$

$$R(\lambda, z_1, z_2) = \int_{z_1}^{z_2} f \cdot \frac{b_b}{a}(\lambda, z) \cdot \frac{dC(\lambda, z)}{dz} \cdot dz, \tag{14}$$

$$C(\lambda, z) = exp\left(-\int_0^z [K_d(\lambda, z\prime) + K_u(\lambda, z\prime)]dz\prime\right), \tag{15}$$

where $R$ is the estimation of irradiance reflectance contribution by the layer from depth $z_1$ to $z_2$, $C$ is the round-trip irradiance attenuation of the signal in the upper layer from 0 to $z$, $K_u(\lambda)$ is the diffuse attenuation coefficient of upwelling irradiance, $dC/dz$ is the weighting function, and $f$ is the coefficient depending on the zenith angle of the Sun, the albedo of single scattering in seawater, and the fraction of molecular scattering of light by suspended particles [55]. In the case of using the formula (Equation (13)), the coefficient $f$ can be neglected and not calculated [20].

It is also possible to calculate the maximum spectral penetration depth of sunlight for remote sensing:

$$Z_{90}^{max} = \max(Z_{90}(\lambda)). \tag{16}$$

However, from the perspective of eddy detection, it is not guaranteed that a significant remotely sensed contrast is formed precisely up to the depth of $Z_{90}^{max}$. A small change in the remotely sensed parameter, resulting from the variability in the 10% of scattered sea radiation that is unaccounted for, may be sufficient for the absolute value of $CNR$ to exceed one. In such cases, the eddy structure should be remotely observed due to the influence of deeper layers beyond $Z_{90}^{max}$. Alternatively, there may be scenarios where initially small contrasts in the analyzed eddy structure led to the eddy structure disappearing at depths smaller than $Z_{90}^{max}$. Therefore, in general, the maximum depth of remotely sensed contrast formation will not necessarily be equal to the $Z_{90}^{max}$.

Here, we provide the following definition:

**Definition 1.** *The maximum depth of the remotely sensed contrast formation of an eddy structure ($Z_{rsE}$) is the greatest depth at which the variability in the eddy structure leads to $|CNR| \geq 1$ for at least one of the remotely estimated parameters.*

In order to estimate the $Z_{rsE}$ value, it is necessary to use in situ bio-optical measurements to account for the real vertical structure of the eddy. Also, it is necessary to use direct numerical modeling methods of light propagation to take into account the influence of each layer of the water column in the formation of $Rrs$. Within the scope of this paper, the algorithm for determining the value of $Z_{rsE}$ involves the following steps:

1. Obtain quasi-synchronous measurements of $Rrs_{sat}(\lambda)$, $Rrs_{ship}(\lambda)$ spectra, and in situ data on the vertical distribution of seawater optical properties within the eddy action region. Ideally, one would require data on the absorption coefficients and volume scattering functions of various optically active constituents, or data from which estimates of these inherent optical properties (IOPs) can be derived [56,57]. In situ data on the concentrations of the main optically active substances in seawater, such as phytoplankton, CDOM, and SSs, can also be utilized. At the very least, only the chl-a concentration is necessary if all other optical characteristics can be estimated from it.
2. Identify the spatial structure of the eddy in $Rrs_{sat}(\lambda)$ and/or $Rrs_{ship}(\lambda)$ data and define areas for signal and background detection.
3. Estimate the statistical noise values of the used $Rrs_{sat}(\lambda)$ satellite data.
4. Set up and validate bio-geo-optical models for direct numerical simulation of $Rrs_{model}(\lambda)$ spectra taking into account vertical profiles of IOPs or optically active substances' concentrations in the study area and by comparison with $Rrs_{ship}(\lambda)$ and/or $Rrs_{sat}(\lambda)$ measurements.
5. From the results of step 4, calculate the $Rrs_{sim}(\lambda)$ spectra under the simulated layer-by-layer removal of the eddy structure in the in situ data.
6. From the results of step 5, calculate the spectral characteristics of $Rrs_{sim}(\lambda)$, $BR_{sim}(\lambda)$, and $chlor\_a_{sim}$.

7. For each characteristic obtained in step 6, calculate the contrast-to-noise ratio $CNR_i$. Then, repeat steps 5 and 6 in 1 m depth increments while the maximum value of all obtained $CNR_i$ values is greater than one.

8. The last depth value calculated in step 7, where the condition $\max(|CNR_i|) \geq 1$ is satisfied, is the maximum depth of the remotely sensed contrast formation of an eddy structure ($Z_{rsE}$).

The implementation of steps 1–3 is described in Sections 2.2–2.4 of the paper. Further, we will discuss the process of step 4 in Section 2.5.1 and step 5 in Section 2.5.2 separately. The description of the algorithm is presented more formally as a flowchart in Figure 5.

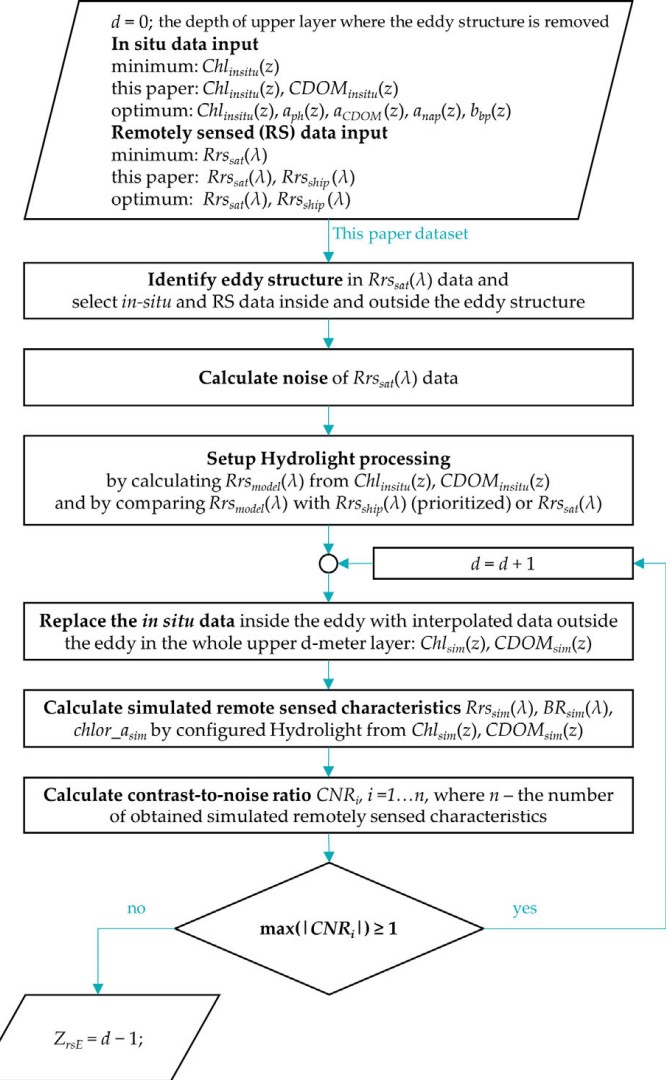

**Figure 5.** Flowchart of the algorithm for determining the maximum depth of a remotely sensed contrast formation of an eddy structure ($Z_{rsE}$). $a_{ph}$—light absorption coefficient by phytoplankton, $a_{CDOM}$—by CDOM, $a_{nap}$—by non-algal particles, $b_{bp}$—light backscattering coefficient by particles.

2.5.1. The Adaptation of Bio-Geo-Optical Models for Numerical Modeling of *Rrs* Spectra

Numerical modeling of $Rrs_{model}(\lambda)$ spectra was performed with Hydrolight-Ecolight 6.0 software [58] using a set of "Case 2" models in which the vertical distribution of chl-a and CDOM concentrations was specified based on in situ measurements of $Chl_{insitu}(z)$ and $CDOM_{insitu}(z)$. The use of the "Case 2" models provided the best results. The quality of

modeling $Rrs_{model}(\lambda)$ was checked by comparison with measured $Rrs_{ship}(\lambda)$ based on the use of the following metric to compare with the results of [59]:

$$\% \delta_{rel} = \frac{100\%}{mean(Rrs_{ship}(\lambda))} \times mean\left(\left|Rrs_{model}(\lambda) - Rrs_{ship}(\lambda)\right|\right). \tag{17}$$

The value of $\%\delta_{rel}$ for the analyzed spectra in the transect through the eddy $ID_{04.02}$ ranged from 6% to 12%, which is considered a good result for this type of comparison. This level of errors is consistent with the results of [59], where the same parameter could reach 20%. Thus, it can be concluded that the settings of the Hydrolight-Ecolight 6.0 software and the input data used are sufficiently valid for simulating the $Rrs(\lambda)$ spectra in the eddy region.

### 2.5.2. Simulation of Layer-by-Layer Removal of the Eddy Structure

For stations located in the periphery or core of the eddy, as illustrated in Figure 3, in situ values of optical characteristics at different depths were replaced by interpolated background values from the surface to the bottom, with an increment of one meter. An example simulation of the equalization of signal values relative to background values for vertical chl-a concentration profiles is presented in Figure 6.

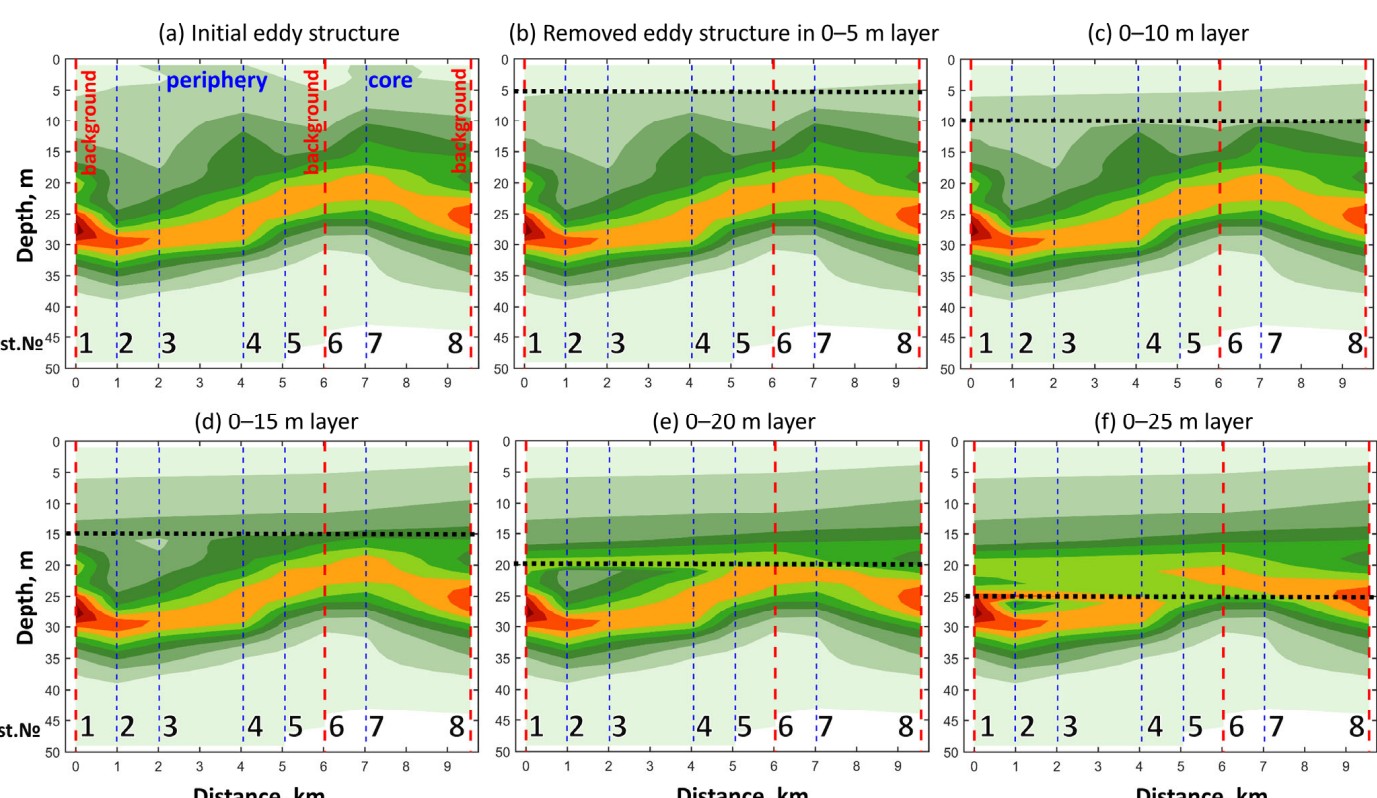

**Figure 6.** Example simulation of layer-by-layer removal of the eddy structure in the $Chl_{insitu}(z)$ concentration profile. Panel (**a**) illustrates the initial structure, then panels (**b**–**f**) show structures where values from "background" stations (st.1, st.6, st.8, red dashed lines) in the sea surface layer are propagated to all "signal" stations (st.2–5, st.7, blue dashed lines) for the following layer thickness ranges: (**b**) 0–5 m; (**c**) 0–10 m; (**d**) 0–15 m; (**e**) 0–20 m; (**f**) 0–25 m. Black dashed lines is the boundary of the processed upper layer. The numbers correspond to the ship's station numbers (st. №).

This procedure was performed for both the chl-a and CDOM concentration profiles. The corresponding resulting vertical profiles $Chl_{sim}(z)$ and $CDOM_{sim}(z)$ were used as input parameters for the adapted Hydrolight procedures to calculate $Rrs_{sim}(\lambda)$.

## 3. Results

### 3.1. Satellite Data Noise Estimation

For the task of determining the most significant contrast characteristic for remote detection of a submesoscale eddy, it is necessary to consider the noise values of the used remotely sensed measurements. Such estimations were carried out in accordance with the methodology outlined in Section 2.4. Table 1 presents the relative noise values $\%noise_{rel}$ obtained for the analyzed satellite dataset.

**Table 1.** Statistical relative noise $\%noise_{rel}$ (Equation (9)) of MODIS-Aqua/-Terra satellite estimates for $Rrs_{sat}(\lambda)$, $BR_{sat}(\lambda)$, $chlor\_a_{sat}$, and $sst$ in the region of the investigated submesoscale eddies from 31 August to 4 September 2009.

| λ, | $\%noise_{rel}$ for $Rrs_{sat}(\lambda)$ | | | | $\%noise_{rel}$ for $BR_{sat}(\lambda) = Rrs_{sat}(\lambda)/Rrs_{sat}(555)$ | | | |
|---|---|---|---|---|---|---|---|---|
| nm | 31 August | 1 September | 2 September | 4 September | 31 August | 1 September | 2 September | 4 September |
| 412 | 24.2% | 6.8% | 7.7% | 9.8% | 25.3% | 7.9% | 9.7% | 11.7% |
| 443 | 7.4% | 4.4% | 3.4% | 4.3% | 7.7% | 4.6% | 5.7% | 8.8% |
| 469 | 5.0% | 3.7% | 2.8% | 2.7% | 5.9% | 4.5% | 5.6% | 8.4% |
| 488 | 3.2% | 1.8% | 2.0% | 2.1% | 5.0% | 3.2% | 4.2% | 7.0% |
| 531 | 2.3% | 2.0% | 1.8% | 2.5% | 2.6% | 2.6% | 2.7% | 4.2% |
| 547 | 2.1% | 2.6% | 2.0% | 2.6% | 1.9% | 2.0% | 1.7% | 3.2% |
| 555 | 2.9% | 3.7% | 3.1% | 3.8% | | | | |
| 645 | 129.5% | 129.7% | 165.3% | 163.6% | 175.7% | 109.6% | 175.4% | 145.3% |
| 667 | 49.8% | 121.1% | 111.6% | 153.8% | 29.7% | 85.1% | 121.7% | 107.7% |
| 678 | 105.5% | 194.0% | 1370.7% | 1917.3% | 33.7% | 211.6% | 200.0% | 200.0% |
| | $\%noise_{rel}$ for $chlor\_a_{sat}$ | | | | $\%noise_{rel}$ for $sst$ | | | |
| | 2.0% | 2.9% | 2.8% | 3.3% | 0.7% | 1.0% | 0.7% | 0.4% |

It can be seen that the spectral range with the lowest relative noise corresponds to approximately 440–560 nm. The relative noise values tend to increase as they get closer to 400 nm. The increase is associated with the reduction in signal due to the additional influence of the CDOM absorption, as well as the increase in the atmospheric correction errors of the satellite data [60,61]. At wavelengths longer than 600 nm, the relative noise values exceed 100% primarily due to the low values of measured *Rrs*.

After normalizing $Rrs_{sat}(\lambda)$ by $Rrs_{sat}(555)$ for $BR_{sat}(\lambda)$ values, the relative noise increases due to error propagation from the two measurements used. In the case of chl-a concentration, $\%noise_{rel}$ is around 3%. And the relative noise for *sst* measurements does not exceed 1%, which can be attributed to the high temperature values, about 20 °C, during the study period.

### 3.2. The Most Significant Contrast Optical Characteristics for Eddy Detection

To estimate the detectability of the hydrodynamic structure of submesoscale eddies, satellite data around all 11 selected eddies shown in Figure 2 were analyzed. For each eddy structure, *CNR* values were calculated from four different sides, under the condition of high-quality data in the background region. As a result, 40 instead of 44 *CNR* values were obtained for each considered optical characteristic for further analysis.

Figure 7a illustrates the number of wins (*n*) for $Rrs_{sat}(\lambda)$ with the highest *CNR* value at each wavelength. Similarly, Figure 7b shows the number of wins for $BR_{sat}(\lambda)$. It can be observed that $Rrs_{sat}(\lambda)$ had the most significant contrast most frequently at the wavelength of 547 nm, and the wavelength of 678 nm also had notable occurrences. For $BR_{sat}(\lambda)$, the best contrast was observed at 678 nm, with a significant number of wins also occurring at 443 nm.

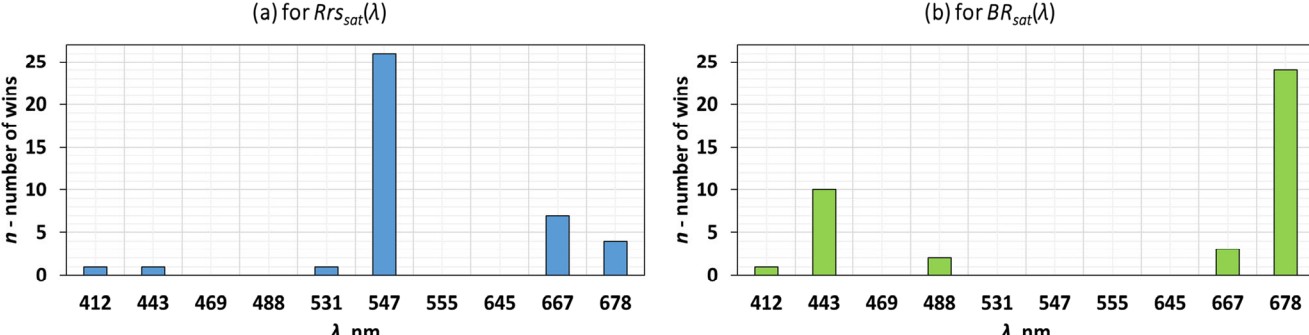

**Figure 7.** Number of wins at which *CNR* was the highest among all MODIS radiometer wavelengths: (**a**) for $Rrs_{sat}(\lambda)$, (**b**) for $BR_{sat}(\lambda)$.

Figure 7 effectively highlights the wavelengths where the best contrasts were observed, but the wavelengths that also have high but not the best *CNR* values are lost. Therefore, we additionally calculated the mean and median *CNR* values for $Rrs_{sat}(\lambda)$ and $BR_{sat}(\lambda)$ for all analyzed eddies. Figure 8 presents the corresponding results and, for comparison, the maximum *CNR* values among all analyzed eddies are given.

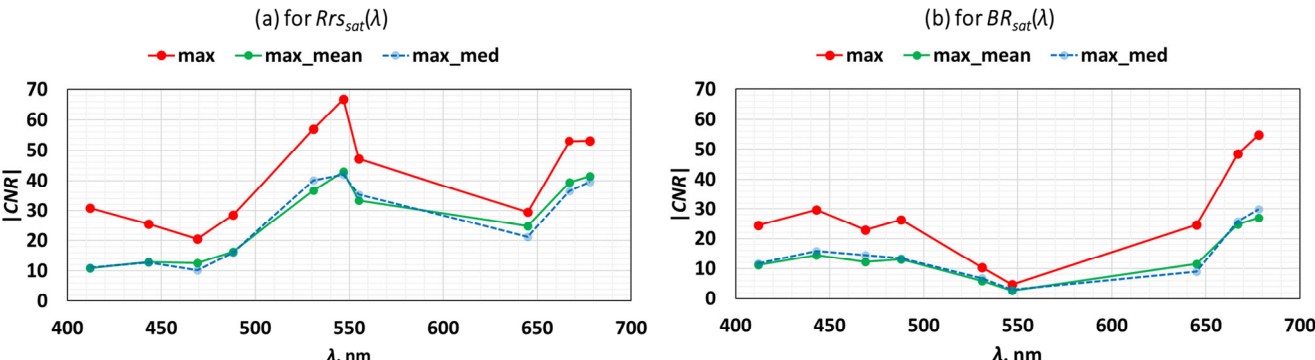

**Figure 8.** Mean *CNR* from the maximum for each eddy (max_mean), median from the maximum *CNR* for each eddy (max_median), and absolute maximum *CNR* for all analyzed eddies (max): (**a**) for $Rrs_{sat}(\lambda)$, (**b**) for $BR_{sat}(\lambda)$ at MODIS radiometer wavelengths.

The results presented in Figure 8 enable the identification of optimal spectral ranges for eddy detection. When using $Rrs_{sat}(\lambda)$ values for the investigated vortices, these are the ranges 520–560 nm and 660–690 nm. And in the case of $BR_{sat}(\lambda)$, the ranges are 440–500 nm and 660–690 nm.

In order to compare the *CNR* values between the different characteristics in the spatial distribution of which eddies can be observed, Figure 9 is plotted. This figure depicts the best contrasts for *chlor_a$_{sat}$*, *sst*, $Rrs_{sat}(547)$, and $BR_{sat}(678)$ for each of the selected eddy structures. It can be seen that the most significant contrast characteristic for submesoscale eddies in the study area is the *chlor_a$_{sat}$*. And the least significant *CNR* characteristic is the *sst*.

Based on the analysis conducted, it can be preliminarily concluded that the redistribution of optically active substances concentrations and changes in their vertical stratification within submesoscale eddies lead to a set of contrasting characteristics in the *Rrs* spectra and in the results of their processing. This finding enables the selection of parameters that are most effective for remotely detecting submesoscale eddies using passive optical sensing methods.

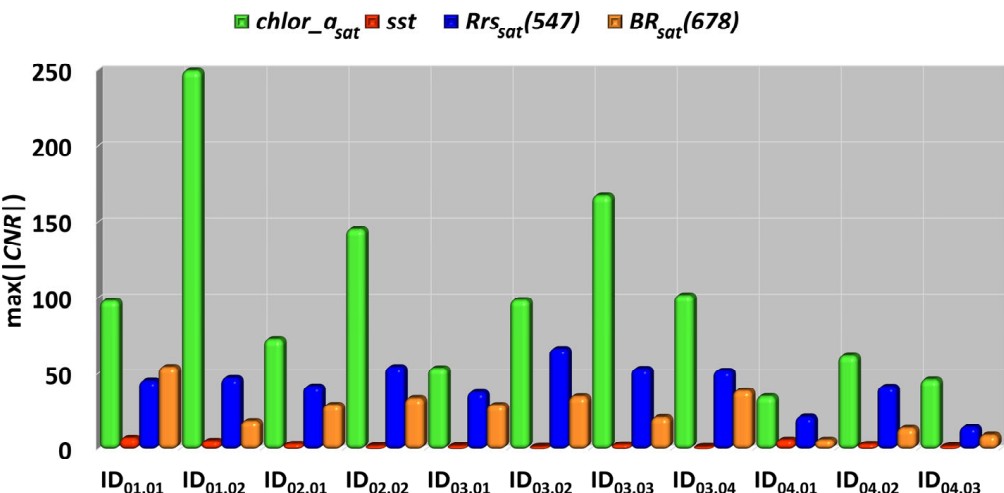

**Figure 9.** Comparison of the maximum absolute *CNR* values for satellite estimations of *chlor_a$_{sat}$*, *sst*, *Rrs$_{sat}$*(547), and *BR$_{sat}$*(678) for the individual eddies shown in Figure 2.

### 3.3. The Maximum Depth of a Remotely Sensed Contrast Formation of an Eddy Structure

According to the methods described in Section 2.5, a dataset comprising synchronized in situ and remote measurements is necessary. Such a dataset was acquired in the region of eddy ID$_{04.02}$ on 4 September 2009. Section 3.3.1 will present the results of the comparison between in situ and remote measurements. In Section 3.3.2, the maximum depth of a remotely sensed contrast formation of an eddy structure ($Z_{rsE}$) will be calculated. Finally, in Section 3.3.3, the obtained depth value will be compared with the sunlight penetration depth for remote sensing ($Z_{90}(\lambda)$).

#### 3.3.1. Comparison of In Situ and Remote Sensing Data in the Area near Eddy ID$_{04.02}$

Figure 10 shows a comparison between satellite data of *sst* and *chlor_a$_{sat}$* with data obtained from ship-based measurements of $T_{flow}$ and $Chl_{flow}$ at the intersection of eddy structures ID$_{04.01}$ and ID$_{04.02}$. Qualitatively, the structures in the satellite and ship data coincide. In the case of structure ID$_{04.01}$, the presence of the eddy is evident through a decrease in temperature and an increase in chl-a concentration. This can be observed in the transect within the latitude range of ~42.46–42.55°N. On the other hand, for structure ID$_{04.02}$, the eddy is not strongly manifested in the temperature measurements. However, in the chl-a concentration fields, increased values are observed within the eddy core region near latitude 42.37°N and in the eddy periphery region, where two local maxima are observed near latitudes 42.40°N and 42.42°N. Additionally, Figure 10d indicates the locations of ship stations made along the transect under consideration.

Figure 11 displays a set of in situ measurements of vertical profiles of temperature $T_{insitu}$ and salinity $S_{insitu}$ conducted at the intersection of structure ID$_{04.02}$. The station numbers correspond to the scheme presented in Figures 1 and 3. It is evident that, similar to the $T_{flow}$ and *sst*, the variability in the water temperature in the near-surface 0–5 m layer of the sea is minimal. Conversely, salinity measurements in this layer exhibit significant variations, ranging from approximately 31.5 to 32.5 PSU. This variation is attributed to the influence of the Tumen River waters within the eddy structure. Lower salinity values are observed at stations 2–5 (eddy periphery) and station 7 (eddy core).

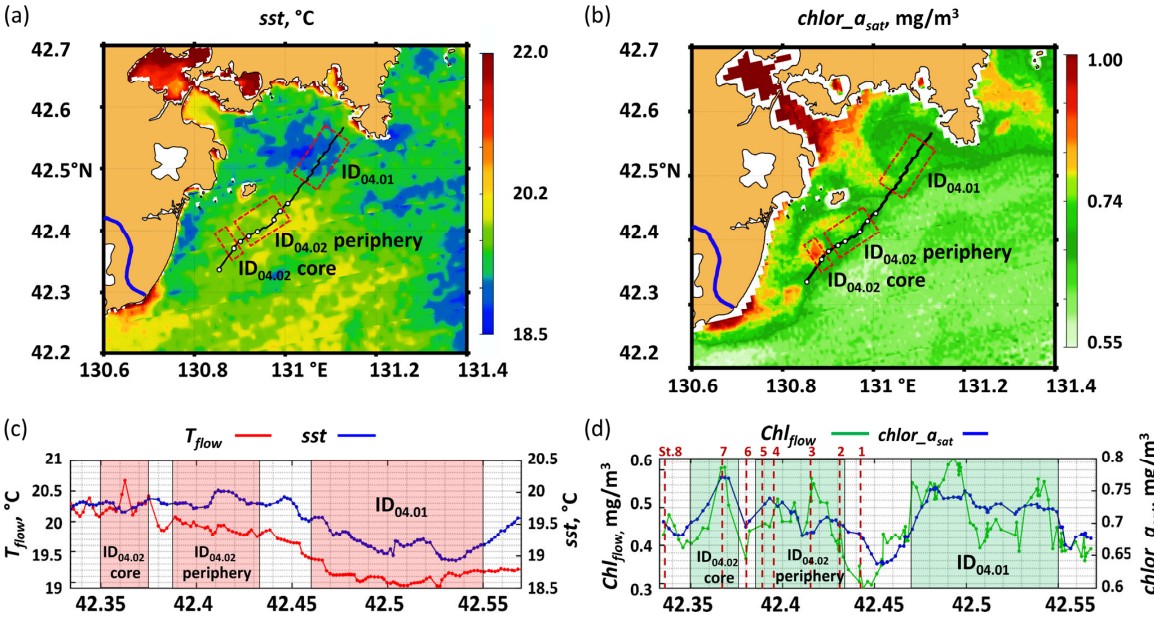

**Figure 10.** MODIS/Aqua satellite images, (**a**,**b**) for 4 September 2009 in the area of two submesoscale eddies ID$_{04.01}$ and ID$_{04.02}$ and their corresponding plots of spatial variability of the analyzed parameters along the ship transect, (**c**,**d**). Panels (**a**,**c**) provide data for *sst* and $T_{flow}$ measurements; panels (**b**,**d**) provide data for *chlor_a$_{sat}$* and *Chl$_{flow}$* measurements. The red and green rectangles in (**c**,**d**) correspond to the eddy boundaries of ID$_{04.01}$ and ID$_{04.02}$.

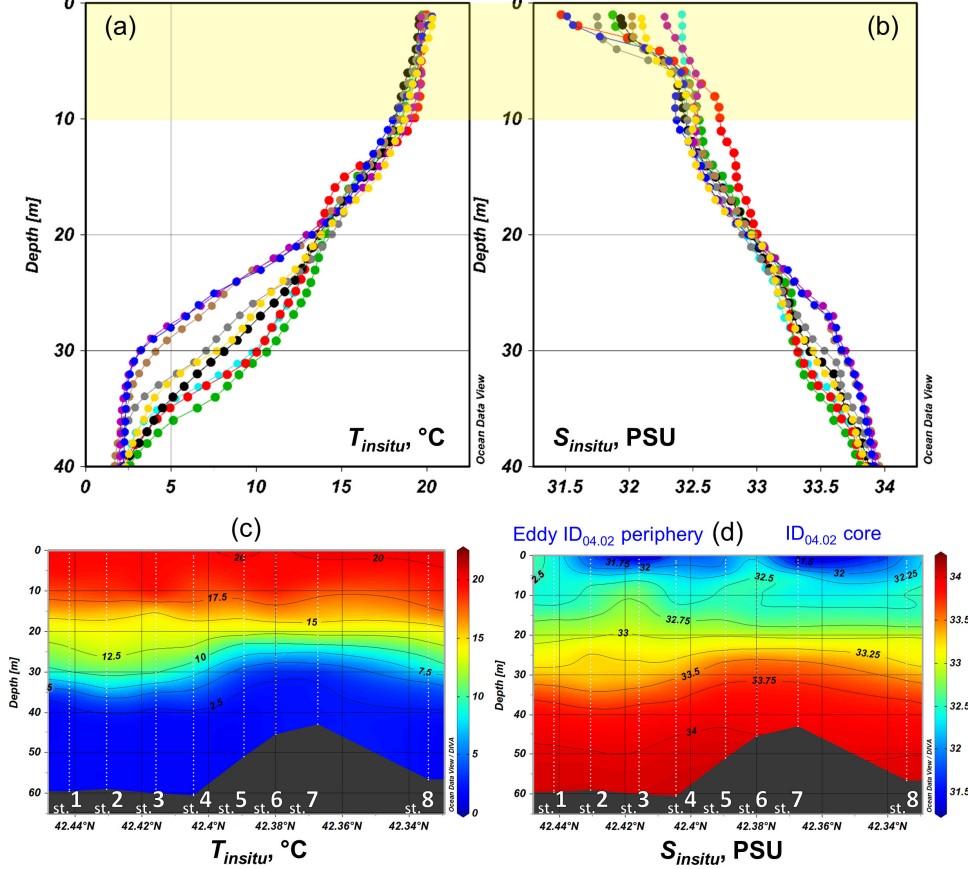

**Figure 11.** In situ vertical profiles of temperature $T_{insitu}$ (**a**) and salinity $S_{insitu}$ (**b**) obtained at the stations located at the intersection of submesoscale eddy ID$_{04.02}$ on 4 September 2009 (**c**,**d**). The colors of the dots in (**a**,**b**) correspond to the colors of the dots in the map shown in Figure 1.

The corresponding measurements of the in situ vertical structure of bio-optical charac­teristics $Chl_{insitu}$ and $CDOM_{insitu}$ in the region of eddy $ID_{04.02}$ are presented in Figure 12. Similar to the salinity measurements, increased concentrations of chl-a and CDOM are observed at stations 2–5, located at the periphery of the eddy, as well as at station 7 in the core of the eddy, within the near-surface layer of 0–5 m. This increase is attributed to the transport of plumes from the Tumen River waters [6]. Additionally, the eddy action at stations 4 and 7 leads to the rise in isolines for both chl-a and CDOM concentrations.

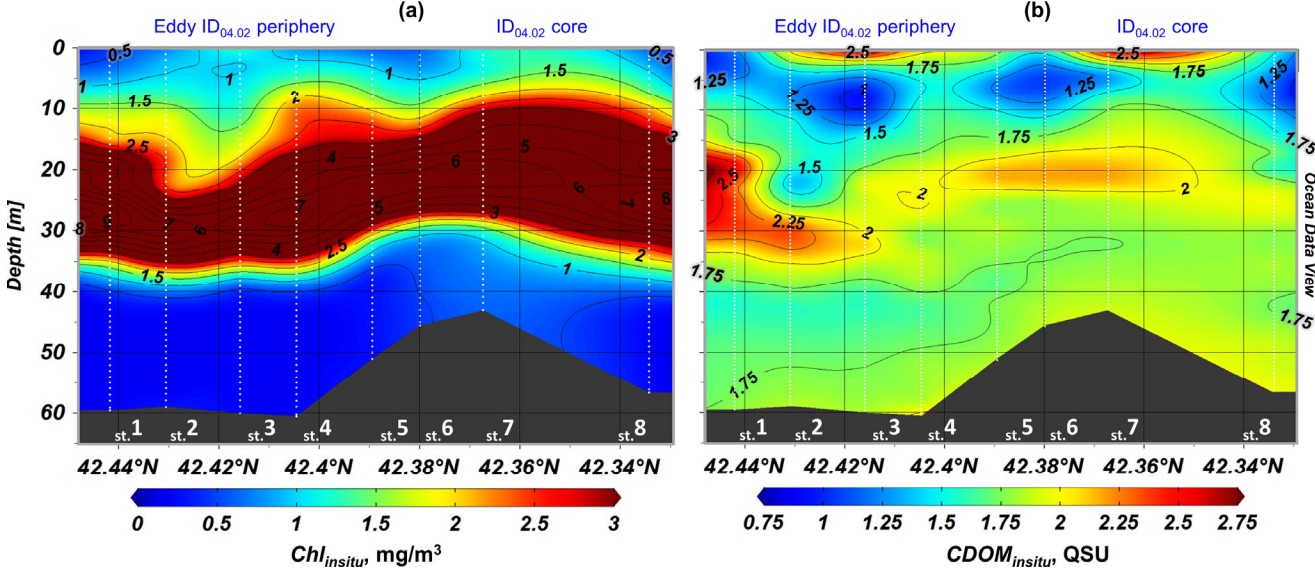

**Figure 12.** Vertical transect of in situ measurements of chl-a $Chl_{insitu}$ (**a**) and CDOM $CDOM_{insitu}$ (**b**) concentrations obtained at the stations located at the intersection of submesoscale eddy $ID_{04.02}$ on 4 September 2009.

Using the data presented in Figure 12 as an input to the Hydrolight bio-geo-optical model set, and comparing the results of numerical simulations of $Rrs_{model}(\lambda)$ spectra with the results of real $Rrs_{model}(\lambda)$ measurements from aboard the ship, the optimal regional parameters for the direct numerical calculation of $Rrs$ as a function of input in situ data on the vertical distribution of chl-a and CDOM concentrations were selected. It was found that the optimal results are obtained by the models for waters of the second optical type, where the influence of additional CDOM sources not related to the activity of phytoplankton communities is significant.

A total of eight pairs of $Rrs_{ship}(\lambda)$ and $Rrs_{model}(\lambda)$ spectra were compared, and the corresponding statistical metrics are presented in Table 2. It is important to note that the measurements of $Rrs_{ship}(\lambda)$ and the vertical profiles of $Chl_{insitu}(z)$ and $CDOM_{insitu}(z)$ were conducted close in time and coordinates but not with a perfect match. The time difference between measurements could range from 5 to 10 min. Within this time frame, the ship could drift several hundred meters away within the submesoscale eddy structure, leading to additional uncertainties between in situ and remote measurements due to the relatively small size of the eddies. Hence, the $Rrs_{model}(\lambda)$ data were used for further analysis to closely match the in situ structure presented in Figure 12.

**Table 2.** Statistical metric $\%\delta_{rel}$ calculated according to a formula (Equation (17)) to compare measured spectra from shipboard $Rrs_{ship}(\lambda)$ and modeled by Hydrolight-Ecolight 6.0 software with regional settings $Rrs_{model}(\lambda)$.

|  | st.1 | st.2 | st.3 | st.4 | st.5 | st.6 | st.7 | st.8 |
|---|---|---|---|---|---|---|---|---|
| $\%\delta_{rel}$ | 11.5% | 6.7% | 11.7% | 6.7% | 10.1% | 7.6% | 5.7% | 10.4% |

3.3.2. Estimation of the Maximum Depth of Remotely Sensed Contrast Formation

First, we analyze the variations in remotely sensed optical characteristics in the region of submesoscale eddy $ID_{04.02}$ using validated model data $Rrs_{model}(\lambda)$ to identify the largest contrasts appearing in the eddy structure at its intersection, in accordance with Figures 10–12. Figure 13 presents the variability in $Rrs_{model}(\lambda)$ by station at wavelengths 443 and 490, as well as the variability in $BR_{model}(443)$.

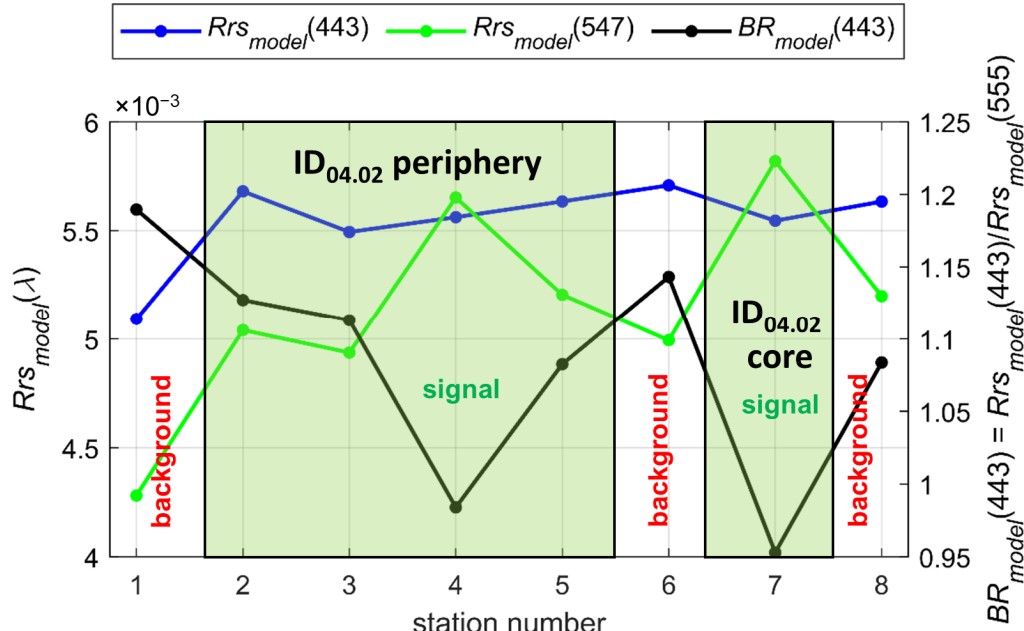

**Figure 13.** Variability in $Rrs_{model}(443)$, $Rrs_{model}(547)$, and $BR_{model}(443)$ values on a transect through submesoscale eddy $ID_{04.02}$.

The eddy structure is well manifested in $Rrs_{model}(547)$ primarily due to the increased variability in the light backscattering in the eddy water, resulting in a positive contrast. And, for $Rrs_{model}(443)$, there are noticeable local decreases in both the periphery and core regions of eddy. These decreases can be attributed to the predominance of absorption variability caused by the high content of optically active seawater substances in the Tumen River waters transported by the eddy. Nevertheless, the overall structure of the eddy is not clearly observed in the $Rrs_{model}(443)$ data. To achieve a clearer depiction of the structure and a better understanding of its relationship with light absorption by phytoplankton and CDOM, it is necessary to analyze the band-ratio $BR_{model}(443)$. As can be seen in Figure 13, this ratio exhibits a negative contrast when crossing the eddy structure.

Figure 13 demonstrates that the largest contrast of the considered characteristics is observed between stations st.4 and st.1, st.4 and st.6, st.7 and st.6, and st.7 and st.8. For comparison, Figure 14 displays the calculated values of the highest absolute contrast-to-noise ratio values, $CNR_i$, for all $Rrs_{model}(\lambda)$, all $BR_{model}(\lambda)$, and for $chlor\_a_{model}$ between all stations located in the signal and background area. The noise values used in the $CNR_i$ calculation by the formula (Equation (8)) were taken from the analysis of satellite data on 4 September 2009, so that the results obtained were closer to the applicability for satellite data.

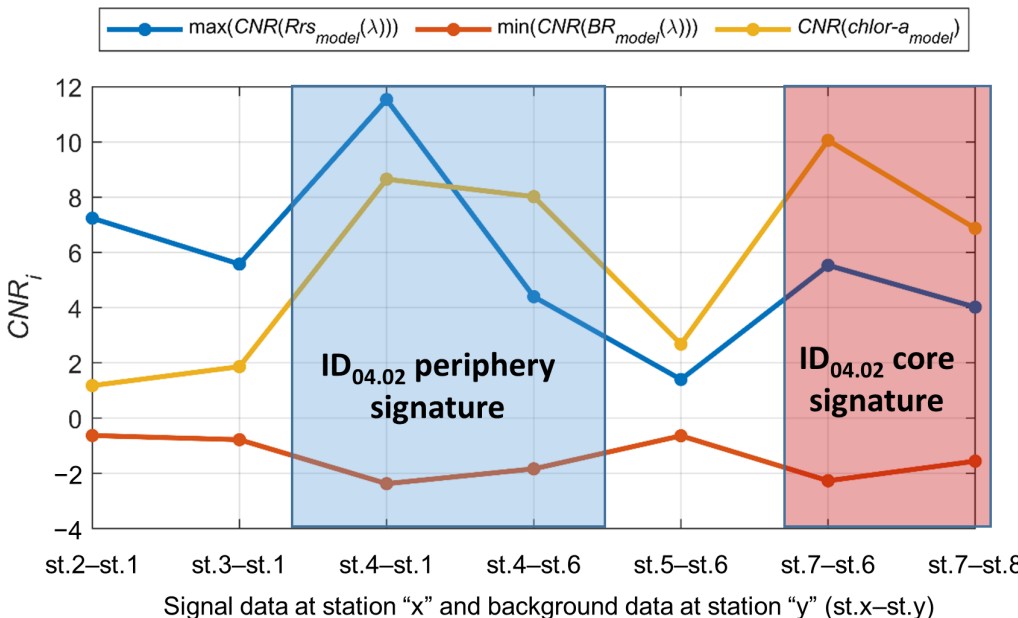

**Figure 14.** Contrast-to-noise ratio values, $CNR_i$, occurring between a set of the $i$-th number of remotely sensed characteristics modeled at transect through submesoscale eddy $ID_{04.02}$. The blue rectangle marks the contrasts characterizing the presence of the eddy periphery. The red rectangle marks contrasts indicating the eddy core manifestation.

The presence of significant contrasts between the data at both station pairs st.4–st.1 and st.4–st.6 indicates that the eddy periphery can be remotely detected. Similarly, the presence of simultaneously significant contrasts at pairs st.7–st.6 and st.7–st.8 suggests that the eddy core can be detected. Furthermore, in order for the eddy structure to be accurately determined, all contrasts should have the same sign. If any of the contrasts in pairs st.1–st.4 or st.4–st.6 are not significant or change sign, it becomes impossible to determine the structure of the eddy periphery. Likewise, if any of the contrasts in st.7–st.6 or st.7–st.8 are not significant or they change sign, it becomes impossible to determine the structure of the eddy core and identify the eddy as a whole.

Figure 15 analyzes the changes in max($|CNR_i|$) values for selected station pairs during the layer-by-layer removal of the eddy structure in the sea's surface layer. This procedure is described in Section 2.5.2 and illustrated in Figure 6. The $CNR_i$ values are calculated using simulated $Rrs_{sim}(\lambda)$, $BR_{sim}(\lambda)$, and $chlor\_a_{sim}$ values obtained from Hydrolight-Ecolight 6.0 software. The input data for these calculations include the modified vertical profiles of $Chl_{sim}(z)$ and $CDOM_{sim}(z)$, applied layer by layer with a 1 m increment. Similar to the $CNR_i$ values presented in Figure 14, the noise values used in this analysis were obtained from satellite data collected on 4 September 2009.

The analysis of max($|CNR_i|$) changes shows that both the eddy periphery and the eddy core lose their structure in the remote sensing data when the depth reaches 7 m. Therefore, the maximum depth of the remotely sensed contrast formation in the considered vertical structure of submesoscale eddy $ID_{04.02}$ ($Z_{rsE}$) was 6 m, in accordance with the condition (Equation (11)) in Section 2.5. This depth value is critical for the formation of both the eddy periphery contrast and the eddy core contrast in the ocean color spectral data.

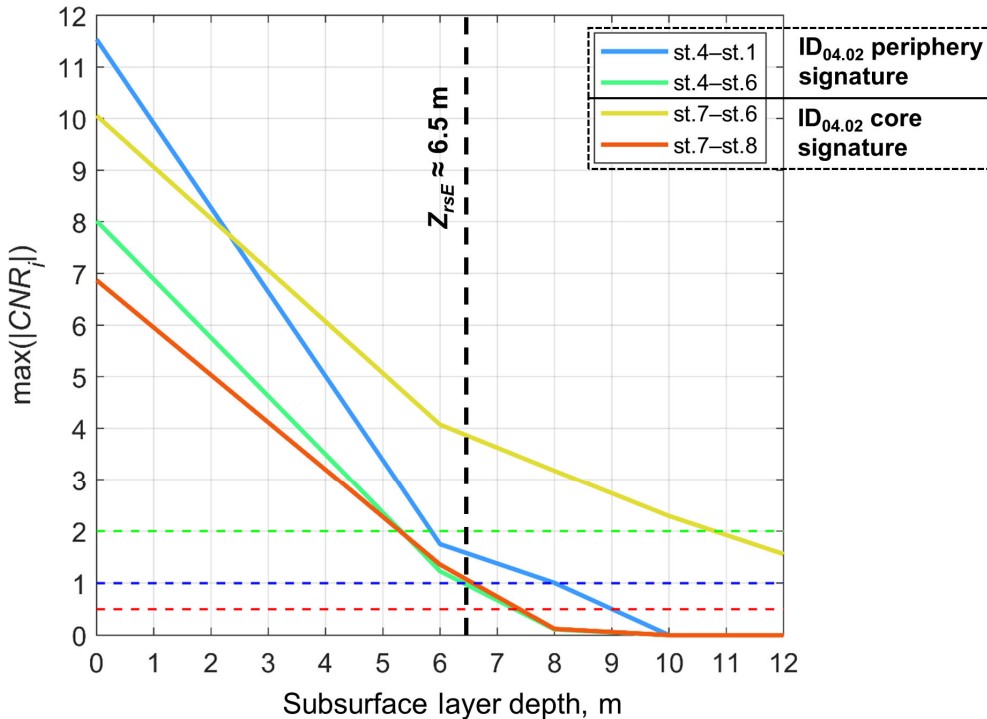

**Figure 15.** Variations in the maximum absolute value of $CNR_i$ among the remotely sensed characteristics, simulated for the case of layer-by-layer structure removal from the bio-optical characteristics' profiles, according to the methodology presented in Figures 5 and 6. Different line colors correspond to different pairs of analyzed stations, according to the legend presented.

### 3.3.3. Comparison with Maximum Sunlight Penetration Depth for Remote Sensing

An additional question is how the depth $Z_{rsE}$ value relates to the estimates of the maximum sunlight penetration depth $Z_{90}^{max}$. With the availability of the regional tuned Hydrolight-Ecolight 6.0 software, it is possible to use an equation (Equation (13)) to determine the maximum sunlight penetration depth. Figure 16a displays the results of the $Z_{90}(\lambda)$ calculation for all stations made through the eddy structure $ID_{04.02}$. Figure 16b presents the corresponding estimates of $Z_{rsE}$ and $Z_{90}^{max}$ plotted along the station number.

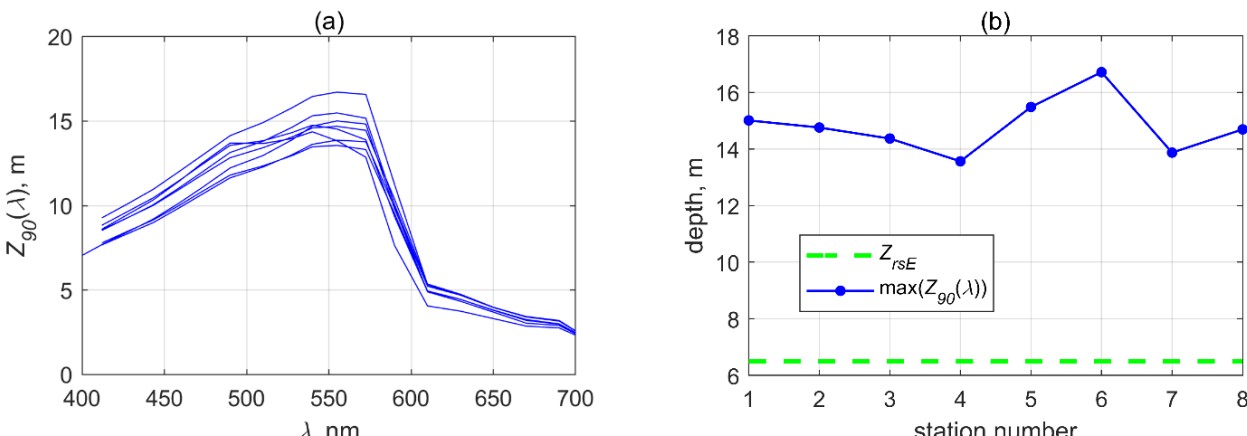

**Figure 16.** (**a**) Wavelength dependencies of $Z_{90}(\lambda)$ calculated for all stations through eddy structure $ID_{04.02}$. (**b**) Comparison of the maximum depth of remotely sensed contrast formation ($Z_{rsE}$) and the maximum depth of sunlight penetration for remote sensing $Z_{90}^{max}$ for considered vertical structures in the eddy $ID_{04.02}$.

The analysis presented in Figure 16 indicates that $Z_{90}^{max} > Z_{rsE}$ for all stations through eddy ID$_{04.02}$. The maximum depths of sunlight penetration for remote sensing are observed in the spectral range of 500–580 nm, where they exceed 12 m and can reach values of approximately 17 m. In the range of 400–480 nm, the depths of $Z_{90}(\lambda)$ vary between approximately 7 and 13 m. And, in the spectral range above 600 nm, $Z_{90}(\lambda)$ sharply decreases and is around 3–4 m.

The variation in $Z_{90}^{max}$ along the ship station numbers shown in Figure 16b corresponds to the eddy structure displayed in Figure 13. It is logical that in the eddy structure, the maximum penetration depth of sunlight for remote sensing is lower than in the region outside the eddy. This is because the waters of the studied eddy contain a large amount of phytoplankton, CDOM, and SSs. In [62], a similar result was obtained for the depth of the photic zone, which decreased inside the cyclonic eddy structure.

## 4. Discussion

### 4.1. Characteristics of the Contrasts

The high contrast for $Rrs_{sat}(547)$ can be attributed to several factors. Firstly, the 547 nm wavelength is located within the transparency window of the seawater, as shown in Figure 16a. In the study area, the transparency window is shifted towards longer wavelengths due to the influence of CDOM and phytoplankton absorption in the blue spectral region [63].

Secondly, 547 nm is within the range that has low errors of atmospheric correction of satellite ocean color data [19,60,61]. And in general, the noise relative to the measured *Rrs* signal at these wavelengths is one of the lowest, as indicated in Table 1.

Thirdly, the increase in $Rrs_{sat}(547)$ values observed in the eddy area (Figures 13 and 14) suggests that the variability in light backscattering is greater than the variability in light absorption at this specific wavelength as it should be in theory [64]. This effect is enhanced by the large number of phytoplankton cells and mineral suspended solids in the eddy waters.

The sometimes-appearing good contrast at 678 nm for both $Rrs_{sat}(\lambda)$ and $BR_{sat}(\lambda)$ is due to the influence of solar-induced chl-a fluorescence (FLH—Fluorescence Line Height). This can be considered as a regional feature because the FLH signal is often low or insignificant [65,66]. However, this finding is important as it demonstrates the usefulness of analyzing the values of $Rrs_{sat}(678)$, $BR_{sat}(678)$, and potentially FLH in satellite-based studies of eddy structures containing high concentrations of chl-a. From the perspective of eddy detection, it is not necessary to use only FLH units. It is sufficient to analyze $Rrs_{sat}(678)$, because it will contain the solar-induced fluorescence of phytoplankton cells within the eddy structure.

Also, a large number of the best *CNRs* for $BR_{sat}(\lambda)$ were observed at the wavelength of 443 nm. This is due to the joint influence of phytoplankton and CDOM absorption [67]. This feature is also good for studying eddy structures because it utilizes the influence of two optically active constituents at once and the relative errors of atmospheric correction of satellite data are still acceptable.

But in general, the significance of the contrasts in the $BR_{sat}(\lambda) = Rrs_{sat}(\lambda)/Rrs_{sat}(555)$ data is lower compared to the $Rrs_{sat}(\lambda)$ data except for some spectral bands and cases. This is attributed to the increase in variance resulting from the division operation and error propagation. This can be seen from the results presented in Table 1. However, normalizing $Rrs_{sat}(\lambda)$ to $Rrs_{sat}(555)$ may be useful in terms of partially reducing the influence of atmospheric correction errors or reducing the effects of sea foam, sun glints, or another weakly wavelength-dependent factor [18,19], and thus a clear eddy structure may be better observed in $BR_{sat}(\lambda)$ than in $Rrs_{sat}(\lambda)$.

It is important to note an additional factor as to why contrasts in the raw $Rrs_{sat}(\lambda)$ data should be used with caution and why $BR_{sat}(\lambda)$ could be useful, especially in the blue wavelength range below 490 nm. This is because *Rrs* is influenced by two competing processes: backscattering and the absorption of individual optically active constituents [64].

And the range below 490 nm will be sensitive to the variability of both of these features in the case of strongly absorbing and strongly scattering substances, which may interfere with the observation of the eddy structure. For example, Figure 13 shows that there is no clear complete eddy structure in the $Rrs_{model}(443)$ data. It is possible to identify the core of the eddy, but the periphery is already lost. At the same time, in $Rrs_{model}(547)$, the structure of eddy $ID_{04.02}$ is fully visible, and it is one of the most significant among all the analyzed characteristics.

It is also necessary to consider the result that the contrasts for $chlor\_a_{sat}$ are more significant than for both $Rrs_{sat}(\lambda)$ and $BR_{sat}(\lambda)$. This is because the relative errors for determining background $chlor\_a_{sat}$ values are smaller than the relative errors for determining background $Rrs_{sat}(\lambda)$ and $BR_{sat}(\lambda)$ values (see Table 1), despite the fact that $chlor\_a_{sat}$ is calculated from $Rrs_{sat}(\lambda)$ and $BR_{sat}(\lambda)$ using a formula (Equation (3)). This effect may be due to the use of a band-difference CI algorithm for chl-a concentrations less than 0.35 mg/m$^3$ (Equations (1) and (3)), which can reduce errors associated with imperfect atmospheric corrections (including sun glint and whitecap corrections) [18]. Such low concentrations of less than 0.35 mg/m$^3$ were the background concentrations in most cases when analyzing the studied chain of submesoscale eddies. It is also shown in [18] that the CI algorithm allows for better identification of eddies compared to using only the band-ratio OCx approach.

The weakest manifestation of the investigated eddies was observed in temperature contrasts compared to almost any optical characteristics. This can be considered as a regional and seasonal result, because the opposite examples are also widespread, when the eddy manifests itself more strongly in temperature contrasts than in bio-optical contrasts [23]. During the study period, the temperature of the upper quasi-homogeneous layer is well warmed and mixed in the analyzed region. At the same time, the temperature of the waters of the Tumen River does not differ much from the temperature of the surrounding marine waters. The temperature difference between the stations does not exceed 0.7 °C.

Thus, the following optimal spectral ranges for detecting the studied eddies can be identified: (1) 440–500 nm in the case of $BR_{sat}(\lambda)$ analysis, primarily due to the influence of phytoplankton and CDOM absorption; (2) 520–560 nm for $Rrs_{sat}(\lambda)$ values due to the location within the transparency window and due to more variable light backscattering coefficients compared to the absorption coefficients in this spectral range; and (3) 660–690 nm, where the additional contribution of solar-induced chl-a fluorescence is observed in the case of high phytoplankton content in the eddy waters.

### 4.2. Sunlight Penetration Depth and the Maximum Depth of Remotely Sensed Contrast Formation

It is important to consider that the different optical characteristics analyzed in this study are formed within different thicknesses of the sea surface layer, depending on the wavelength used. And the light penetration depths inside and outside the eddy differ (see Figure 16). As a result, this can lead to additional contrasts in remote sensing ocean color data due to the fact that inside the eddy, the features are formed in one thickness of the near-surface layer, while outside the eddy, they are formed within another thickness of the layer.

For the investigated eddies, the spectral range ~500–570 nm of the largest sunlight penetration depths $Z_{90}(\lambda)$ approximately coincided with the spectral range of the maximum contrast-to-noise ratio values for $Rrs(\lambda)$ (Figures 8 and 16). Based on the data presented, it is difficult to determine whether this observation can be considered a general rule or not. It is necessary to conduct a larger number of similar studies with different eddy structures.

The estimated value of the maximum depth of contrast formation ($Z_{rsE}$) of the $ID_{04.02}$ eddy structure in the ocean color spectral data was 6 m. This value approximately coincides with the thickness of the upper freshened layer including the waters of the Tumen River enriched with a high content of phytoplankton, CDOM, and SSs (Figures 11 and 12). At the same time, the obtained estimate of $Z_{rsE}$ is significantly lower than the estimate of the maximum sunlight penetration depth for remote sensing $Z_{90}^{max}$, which amounted to 14–17 m. This may be due to the following two reasons.

The first of the reasons is that the determination of $Z_{rsE}$ depends on the quality of the remote measurements, unlike $Z_{90}^{max}$. If the noise of the registered remote sensing data is higher, $Z_{rsE}$ will decrease according to Equations (8) and (11), and vice versa. The $Z_{90}^{max}$ definition does not include consideration of the quality of the measured characteristics [20,54].

The second reason is the dependence of $Z_{rsE}$ on the contrast value of the eddy structure. The $Z_{90}$ depth for *Rrs* formation can be large enough, but when we talk about the contrast, we analyze the difference between the two values calculated from the initial *Rrs* values inside the eddy and outside the eddy. And this difference can become insignificant much earlier than when the $Z_{90}$ depth is reached.

It is highly probable that a smaller value of $Z_{rsE}$ compared to $Z_{90}^{max}$ will be a typical situation in most cases. This is due to the use of the difference of characteristics inside and outside the eddy for estimating $Z_{rsE}$. However, we cannot immediately exclude cases where $Z_{rsE}$ will be larger than $Z_{90}^{max}$. This behavior can be expected when certain conditions are combined: for instance, if there is very high contrast in the hydrodynamic structure of the eddy in the 10% of the unaccounted for scattered radiation for remote sensing, if the accuracy of remote sensing measurements is high, and if the spatial resolution of remote sensing measurements is much smaller than the linear size of the eddy, so that the corresponding eddy structure includes many nearby pixels with a changed remotely sensed signal.

## 5. Conclusions

The redistribution of the content of optically active substances in seawater and changes in their vertical stratification in submesoscale eddies leads to a set of contrasting characteristics in the spectra of the remote sensing reflectance and in the results of their processing. It enables the selection of specific parameters for the most effective detection of submesoscale eddies in remotely sensed ocean color spectral data.

The spectral range of the most significant contrast depends on many factors: the predominant contribution of optically active substances that can lead to light absorption or scattering or to solar-induced fluorescence; the penetration depth of sunlight for remote sensing inside and outside the eddy; and the statistical noise of remote measurements.

When selecting the best contrast characteristic of a submesoscale eddy, not only the contrast but also the data quality should be considered. In each specific case, it is necessary to analyze and choose the best method to solve the problem.

The following specific findings are obtained for the investigated eddies:

- The best parameter for detecting investigated eddies in the satellite ocean color data was the concentration of chl-a, calculated using the band-difference CI algorithm for low chl-a concentrations and band-ratio OC algorithm for high chl-concentrations. The application of such bio-optical algorithms had the effect of enhancing the significance of contrast due to reduction in the noise at background values of chl-a concentrations less than 0.35 mg/m$^3$. At the same time, the weakest contrasts were in *sst* data due to the similar heating of sea and river waters involved in the eddy.
- The best contrast-to-noise ratio according to *Rrs* spectra for the investigated eddies is achieved at a wavelength of 547 nm near the seawater maximum transparency and in the spectral region of low relative errors of satellite ocean color data. The increased contrast values at this wavelength can be related both to the greater variability in the light backscattering by optically active substances contained in the eddy and due to the different penetration depth of solar radiation inside and outside the eddy.
- The *Rrs*(678) value and associated products can be highly significant characteristics for eddy detection if the eddy waters are reached by phytoplankton. The corresponding contrasts inside and outside the eddy will depend on the influence of both the variability of the backscattering and absorption signals of optically active substances and the sun-induced fluorescence signals of phytoplankton cells in the eddy structure.
- When the *Rrs* spectra are normalized to *Rrs*(555), the eddy contrasts become less significant, but the eddy structure itself can be clearer by reducing the influence of

non-hydrodynamic effects on the measurement results. A good characteristic in this case for the calculation of *CNR* is $Rrs(443)/Rrs(555)$, variations of which primary depend on the combined effect of phytoplankton and CDOM absorption, and relative statistical noise level is acceptable.

- The maximum depth of contrast formation ($Z_{rsE}$) of the considered eddy vertical structure $ID_{04.02}$ was 6 m, which is significantly less than the maximum spectral penetration depth of solar radiation for remote sensing $Z_{90}^{max}$, which was in the range of 14–17 m in the eddy area. Probably, a smaller value of $Z_{rsE}$ compared to $Z_{90}^{max}$ can be a typical situation not only for the considered eddy.

The results obtained can be used to determine the hydro-optical characteristics that provide the best contrast for detecting different types of hydrodynamic structures from spectral remote sensing reflectance measurements.

The method of determining the maximum depth of contrast formation improves the interpretation of remote spectral ocean color data. This method considers the vertical variability in optically active substances in seawater and allows for determining the thickness of the surface layer of the sea where hydrodynamic structures can be remotely detected in the visible spectrum. The method also takes into account the quality of remote sensing measurements.

**Author Contributions:** Conceptualization, N.A.L. and P.A.S.; Methodology, N.A.L. and P.A.S.; Software, N.A.L. and P.A.S.; Validation, I.A.G.; Formal Analysis, N.A.L.; Investigation, N.A.L. and P.A.S.; Resources, P.A.S.; Data Curation, P.A.S.; Writing—Original Draft Preparation, P.A.S. and N.A.L.; Writing—Review and Editing, P.A.S. and N.A.L.; Visualization, N.A.L. and P.A.S.; Supervision, P.A.S.; Project Administration, P.A.S.; Funding Acquisition, P.A.S. All authors have read and agreed to the published version of the manuscript.

**Funding:** This research was funded by the Russian Science Foundation, grant number 22-27-20099, and by the Government of Primorsky Krai, agreement number 2-N dated 25 May 2022.

**Data Availability Statement:** Publicly available satellite ocean color datasets were analyzed in this study. This data can be found here: "https://oceancolor.gsfc.nasa.gov/ (accessed on 27 November 2023)". The field data presented in this study are available on request from the corresponding author.

**Conflicts of Interest:** The authors declare no conflict of interest.

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
