# Peer review of "Variations and Depth of Formation of Submesoscale Eddy Structures in Satellite Ocean Color Data in the Southwestern Region of the Peter the Great Bay"

_remotesensing, doi:10.3390/rs15235600_

Round 1

Reviewer 1 Report

Comments and Suggestions for Authors

The study in this paper is to determine the most significant contrasts in satellite ocean color data of submesoscale eddy structure, and the corresponding depths where these contrasts are formed in the upper layer of the sea combined with the results of ship surveys.  Some advices are as follows.

²  There are some analyses about how to obtain cha-l concentration. Please provide the reason for recalculating cha-l, becasue there are standard products of cha-l for MODIS. 

²  The title of 2.2 is not accurate. there are many contents about the calculation of cha-l.

²  The SST data used in this study is not introduced in detail.

²  The determination of submesoscale eddy center and boundary is very important in this study. How to obtain the center and boundary as show in Fig.4.

²  Is there an error in the formula (13)? 0.9 equals a expression?

²  The English expression  need to improve. Individual spelling errors, such as “Satrllite” in the Title 3.1, should be corrected.

Comments on the Quality of English Language

The English expression  need to improve.

Author Response

Thank you very much for your work on reviewing our paper. We sincerely appreciate the time you have invested to improve our manuscript. Please see the response to your review in the attached docx file.

Reviewer 2 Report

Comments and Suggestions for Authors

The article titled: Contrasts and Depth of Formation of Submesoscale Eddy 2 Structures in Satellite Ocean Color Data in the Southwestern 3 Region of Peter the Great Bay. The objective of this study is to develop methods to determine the most significant contrasts in satellite-derived ocean color data that arise in the presence of a submesoscale eddy structure, as well as to determine the corresponding depths of the upper layer of the sea where these contrasts are formed. The research was carried out based on the example of the identified submesoscale eddy chain 13 in the water transport zone of the Tumen River in the Sea of Japan/East Sea.

General comments: The manuscript is well written. The results are as stated in the objectives. A great job of data processing is appreciated. Although the manuscript contributes to the study of the presence of a submesoscale eddy structure, it is not related to other factors, both climatic and anthropogenic. However, due to the importance of local studies and the increasing use of satellite imagery as valuable data, I believe that, after review, its publication should be considered.

Specific Comments:

1. I suggest adding more bibliography, the introduction section does not reach 20 citations (suggested for this section).

2. In figure 1 and 4 it is necessary to add missing cartographic elements (wind rose or north indicator, legend, scale).

3. for international readers it is suggested to add in which part of the country the Tummer River and Peter the Great Bay are located, the figure is very specific.

4. bring the values of the statistics to 2 significant figures after the decimal point (revise the whole body of the manuscript).

5. In figure 2 unify the scale of the variables, the range is from 0.38-9.19 mg/m3 in one picture and in another picture a different range.

6. define CDOM the first time the parameter is mentioned (line 174)

7. line 372 change Satrllite for satellite

Author Response

(The authors gave the same response as above.)

Reviewer 3 Report

Comments and Suggestions for Authors

The paper looks interesting, and the work presented here has excellent utilization in tackling various ocean-based applications. However, I suggest authors view general points which are important to be taken care of before final acceptance. 

 1) The synthetic aperture radar (SAR) data could be a viable solution in performing identifying and characterizing the submesoscale eddies [1]. This could be even more efficient in quantifying the smaller size-based submesoscale eddies, too. In recent works, there are a few very interesting analyses reported in the open literature. However, those are completely absent in this presented manuscript. Please discuss them in detail, as these are also primarily based on Satellites and can provide all-weather and day-and-night facilities for acquiring the datasets.

[1] Xia L, Chen G, Chen X, Ge L and Huang B (2022) Submesoscale oceanic eddy detection in SAR images using context and edge association network. Front. Mar. Sci. 9:1023624. doi: 10.3389/fmars.2022.1023624

2) Do you believe that correctly quantifying submesoscale eddies can also be helpful in increasing the accuracy of ocean applications such as oil spills [2] and iceberg detection [3]? If yes, please discuss them in the paper, too, to establish the applicability of the presented work.

[2] A. Kumar, V. Mishra, R. K. Panigrahi and M. Martorella, "Application of Hybrid-Pol SAR in Oil-Spill Detection," in IEEE Geoscience and Remote Sensing Letters, vol. 20, pp. 1-5, 2023, Art no. 4004505, doi: 10.1109/LGRS.2023.3258224

[3] Soldal, I.H.; Dierking, W.; Korosov, A.; Marino, A. Automatic Detection of Small Icebergs in Fast Ice Using Satellite Wide-Swath SAR Images. Remote Sens. 201911, 806. https://doi.org/10.3390/rs11070806.

3) I suggest adding a block diagram explaining the steps followed in the implementation of the dataset, along with the clear preprocessing or calibration steps used in this work.

Other Corrections:

(a) Page 21: In the sentence ". . . if the eddy waters are reach in phytoplankton . . ", replace "reach" with "reached".

(b) Page 20: Replace the sentence ". . . contrast due to reducing of noise. . . "with ". . . contrast due to the reduction of noise . . ."  

(c) Page 2: In the sentence ". . . satellite IR sensing fail to identify. . .  ", replace "fail" with "fails" to gramatically correct it.

Comments on the Quality of English Language

Minor corrections are needed. I have listed a few of them in my detailed comments. However, it is recommended to rectify all the grammatical and typo errors so that the actual meaning of the sentences can be conveyed to the readers of this manuscript. 

Author Response

(The authors gave the same response as above.)

Reviewer 4 Report

Comments and Suggestions for Authors

the paper entitled "Contrasts and Depth of Formation of Submesoscale Eddy Structures in Satellite Ocean Color Data in the Southwestern Region of Peter the Great Bay" described the submesoscale eddy structure using satellite data and observational data.

authors have done good work and the results are noticeable. however, please check the sentences and modify them properly.

for example: Thus, for a complete study of submesoscale eddies using satellite data in the optical range, it is important to use both ocean color and temperature data and to estimate the depths at which these phenomena are visible in remotely sensed data.

the sentence is complex, write it clearly.

check other sentences in the paper.

Author Response

(The authors gave the same response as above.)

Reviewer 5 Report

Comments and Suggestions for Authors

Contrasts and Depth of Formation of Submesoscale Eddy 2 Structures in Satellite Ocean Color Data in the Southwestern 3 Region of Peter the Great Bay by Lipinskaya et al.

This study addressed how ocean color and infrared remote sensing can detect submesoscale once eddies. There are many reasons why this scale of eddies is important for cascading one energy from mesoscale to more smaller scales. Although the focused research regions and periods are quite limited in addressing the generalized ideas, I think this study can be published after revising the following concerns.

Major:

I was very interested in the step-by-step processes to finalize the spatial boundary of the eddies. The authors used noise and contrast-to-noise ratio (CNR) (Equations 7 and 8). I read many times the definition of noise in this study (Page 6). However, I do not understand the physical meaning of it and how the authors define and prove the noise signals from remote sensing signals. The authors need to justify the noise signals to define CNR.

Another major comment is about Figure 10. It shows the vertical structure of T and S.  I thought it should show the submesocale eddy structure without any explanations. However, I do not discriminate eddy signals (even with explanations in ms).   The authors need to provide clearer ideas as to why this T and S can support the vertical structure of the eddy.

Minor

The word, contrasts, in the title and the ms is strange to me. I may be wrong for this expression in this paper, but I like to suggest another proper word if the authors agree. 

The word, satrillite, shoud be Satellite in Line 372.

Author Response

(The authors gave the same response as above.)

Round 2

Reviewer 3 Report

Comments and Suggestions for Authors

I reviewed the initial version of this manuscript and provided my comments. The comments are rectified well in this revised version, and the proper clarifications are provided correspondingly. I am satisfied with the author's response and recommend publishing it.

Reviewer 5 Report

Comments and Suggestions for Authors

I am satisfied with the current revision.